# Robustness Analysis of Video-Language Models Against Visual and Language Perturbations

**Madeline C. Schiappa**
University of Central Florida
madelineschiappa@knights.ucf.edu

**Shruti Vyas**
University of Central Florida
shruti@crcv.ucf.edu

**Hamid Palangi**
Microsoft Research
hpalangi@microsoft.com

**Yogesh S. Rawat**[*]
University of Central Florida
yogesh@crcv.ucf.edu

**Vibhav Vineet**[*]
Microsoft Research
vivineet@microsoft.com

## Abstract

Joint visual and language modeling on large-scale datasets has recently shown good progress in multi-modal tasks when compared to single modal learning. However, robustness of these approaches against real-world perturbations has not been studied. In this work, we perform the first extensive robustness study of video-language models against various real-world perturbations. We focus on text-to-video retrieval and propose two large-scale benchmark datasets, *MSRVTT-P* and *YouCook2-P*, which utilize 90 different visual and 35 different text perturbations. The study reveals some interesting initial findings from the studied models: 1) models are more robust when text is perturbed versus when video is perturbed, 2) models that are pre-trained are more robust than those trained from scratch, 3) models attend more to scene and objects rather than motion and action. We hope this study will serve as a benchmark and guide future research in robust video-language learning. The benchmark introduced in this study along with the code and datasets is available at https://bit.ly/3CNOly4.

## 1   Introduction

Human beings learn different skills sequentially and in a continual manner. Sequential data like video and language are natural forms of input to any intelligent vision system operating in the real world. Robustness of these intelligent systems against real-world distribution shifts is crucial for various applications including autonomous driving [36, 26, 17, 43], medicine [4, 2, 31, 54], robotics [31, 65, 32, 6] and others. In a multimodal setting where both language and video are used, these distribution shifts can occur for a variety of reasons. In video, these can include lighting, camera movement, digital compression, etc. In text, these can include spelling errors, incorrect synonym swapping, bias, etc. These distribution shifts can cause deep learning models to fail when deployed in a real world setting [28, 12, 17].

It is crucial that these models are robust against such distribution shifts for successful deployment. Robustness has been an active topic of research in deep learning. However, most of the effort is directed towards robustness against adversarial attacks [9, 3, 20]. There are some recent efforts on robustness against real-world distribution shifts, but they focus on non-sequential image data [28, 8, 27] and natural language [59] independently. Because video and text are vital sequential inputs for real-world intelligent systems, studying robustness in a multimodal setting is an important step towards developing reliable systems and has never been studied before.

---

[*]The authors contributed equally as supervisors to this paper.

36th Conference on Neural Information Processing Systems (NeurIPS 2022) Track on Datasets and Benchmarks.

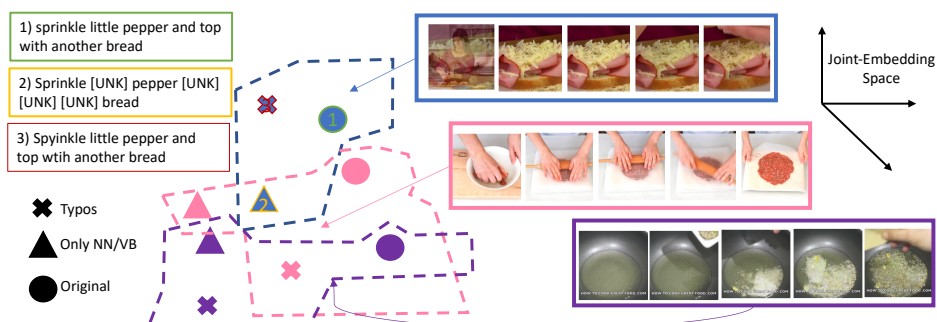

Figure 1: A conceptual diagram of video and text in a joint latent space where the original text (circles) are closer to their paired video compared to text that is perturbed via typos (cross) and the removal of all words except nouns and verbs (triangle). Models are considered robust when the perturbed text is still closest to its respective video. The same should be true if video is perturbed or both are perturbed.

In this work, we perform a large-scale analysis on the robustness of existing multimodal deep learning models for text-to-video retrieval. Text-to-video retrieval provides an important test scenario for a multimodal setting as it evaluates the similarity between video and text embeddings and how their joint-embedding space may vary based on distribution shifts on one or both modalities. There are several questions about existing methods which are unanswered. Are these approaches robust to real-world corruptions in one modality and even both? Do we really need a heavy pre-training strategy for robustness or is training on the target dataset enough? Are the recently introduced transformer-based models better for robustness? Do these approaches utilize temporal modeling? Are these models biased? This study aims to be the first to answer some of these critical questions for video-language deep learning models.

Towards this goal, we present two benchmark datasets to conduct robustness analysis on text-to-video retrieval. We utilize two widely used retrieval datasets MSRVTT [55] and YouCook2 [69] and propose corresponding benchmark datasets, *MSRVTT-P* and *YouCook2-P*. In order to create these benchmarks, we introduce 90 different visual perturbations and 35 textual perturbations.

This study reveals several interesting observations about robustness of video-language models: 1) The studied models are more robust when only text is perturbed as opposed to when only video is perturbed. 2) Model pre-training improves both robustness and performance. 3) Models attend more to object and scene rather than motion and action. We make the following contributions in this study,

- We focus on robustness of video-language approaches against distribution shifts due to spatial/spatio-temporal visual and text perturbations; this problem has not been studied before to the best of our knowledge.
- We provide two large-scale benchmark datasets (MSRVTT-P and YouCook2-P) to conduct robustness analysis on text-to-video retrieval.
- We present an empirical analysis of video-language approaches to study the effect of various perturbations on their performance.

## 2 Related Works

### 2.1 Robustness

**Visual** Most recent works on robustness in the visual domain have focused on real-world distribution shifts as opposed to targeted attacks in the image domain [28, 8, 27, 50, 56]. In [28, 46], authors analyze different image classification models on naturally occurring distribution shifts using ImageNet. While the benchmark study analyzing naturally occurring shifts in [56] demonstrated that data augmentation is not sufficient for robustness, several studies have found that certain data augmentations do improve the robustness of deep learning image models [24, 29, 66]. These data augmentations are often noise related [37, 49, 34] but other transformations such as color or texture have been analyzed as well [24, 68, 13, 29]. These studies have not yet been extended to the video

domain where temporal aspects are also present. Different from these works, this study will provide a benchmark on robustness of models against real-world perturbations in multi-modal settings.

**Text** Research on robustness in the natural-language processing (NLP) field is far more extensive as compared to video. Some works in natural distribution shifts focus on semantic changing of a phrase [23, 51]. In [23], the phrase is altered in small, meaningful ways that change the overall label in order to understand the decision boundaries of models. Similarly, [51] alter text in ways that change the semantic meaning but keep the original text's lexical surface form. Other works inspired by [30] focus on distribution shift based on changes of grammar errors, dialects, speakers, and language [16], different domains [42] and bias [15, 45]. The image robustness research space has inspired many of these studies, but there are vast differences to NLP and vision that make these transfers difficult, such as the discrete vs. continuous search space as explained in [60]. Data augmentation has also been looked at as a method to improve robustness and has shown substantial improvements [22, 19, 11, 29, 10]. These studies have not yet been extended to the multimodal domain where vision is also incorporated. Different from these works, this work will provide a large-scale benchmark on robustness for multimodal models against real-world perturbations.

**Multimodal** Evaluating robustness in multimodal models is more difficult because there are more vectors of attack possible. It is possible to attack the entire model while perturbing only one of the modalities used or a varying amount of the modalities used. Focus on single-modality attacks in [64] investigated the robustness of multimodal neural networks against worst-case (i.e., adversarial) perturbations on a single modality. Looking at multimodal attacks, [58] evaluated audio-visual models by running adversarial attacks on audio, visual, and both modalities. A more general benchmark was proposed in [33] for a variety of modalities. Different to this benchmark, we focus on robustness analysis of the video and text embedding space in great detail. Such studies have not been performed on naturally occurring distribution shifts and have not looked at video-language models specifically, two modalities that are drastically different.

## 2.2 Video-Language Modelling

Multimodal modeling with text and vision has improved since the emergence of both the HowTo100M dataset [39] and transformer architecture [18]. The highest performing models [35, 63, 62, 48, 1, 44] pre-train on the [39] and most use pre-extracted visual features from the original multimodal model from [38] which uses an S3D-G backbone [61]. For learning a joint visual-text space, these models often use a contrastive learning objective between visual and text embeddings [38, 63, 1, 48] while some use an alignment-based objective [35, 62] using masked modeling. Many of the contrastive approaches [63, 38, 48], use a two-branch encoder approach where video has one encoder and text a separate encoder and the objective is to move the two enoder outputs closer to each other in latent space. Some approaches [35, 62, 1] will additionally utilize a cross-encoder before comparing output. This work will provide a greater understanding of these video-language models and their robustness.

## 3 Distribution Shift

Existing research in multimodal learning is mostly focused on training and testing the proposed methods on a benchmark dataset with little to no distribution shift from training to testing samples. While models often use a video encoder that is pre-trained on a very large, noisy dataset, e.g. HowTo100M [39], there is no understanding of how, in a multimodal setting, a distribution shift will affect the joint-embedding space of video and text. To study the effect of distribution shift, we introduce five categories of visual perturbations and seven categories of text perturbations. More details about these perturbations are provided in the Appendix.

## 3.1 Visual Perturbations

First, we extend image-based perturbations from [28] to videos. Next, we add temporal perturbations to address the time dimension and video compression to address video-specific distribution shifts as well as spatio-temporal. The total set of visual perturbations fall into 5 categories: **Noise**, **Blur**, **Temporal**, **Camera** and **Digital**. Each visual perturbation has a severity range from 1 to 5 where the greater the severity, the more challenging and perturbed the video is. Blur, Noise, and Camera

Table 1: Details of self-supervised video-language models used in this study.

| Model | Params | Text Input | Text Encoder | Video Input | Video Encoder |
|---|---|---|---|---|---|
| HowTo100M MIL [38] | 31.2M | Raw | Word2Vec [41] | Raw | S3Dg [61] |
| VideoClip [63] | 177.4M | Raw | BERT [18] | S3D [38] | MLP+Transformer |
| UniVL [35] | 153.7M | Raw | BERT [18] | S3D [38] | Transformer |
| COOT [25] | 7.6M | BERT [18] | Transformer | S3D [38] | Transformer |
| FIT [5] | 180.9M | Raw | BERT [18] | Raw | ViT [21, 7] |

perturbations are all applied frame-by-frame. Noise includes *Impulse*, *Gaussian*, *Shot*, and *Speckle*, Blur includes *Zoom*, *Defocus* and *Motion* and Camera includes *StaticRotate*, *Rotation* and *Translation*.

The Digital and Temporal perturbations are added in order to include distribution shifts specific to video while also perturbing spatially and temporally. **Digital** perturbations are related to compression and video-streaming quality. We evaluted models on *JPEG*, *MPEG1* and *MPEG2*. JPEG is a lossy image compression, MPEG1 compresses video without excessive quality loss and MPEG2 is a lossy compression for video that is similar to MPEG1. **Temporal** perturbations focus on the time dimension in a video and include *Sampling*, *Reverse Sampling*, *Jumbling*, *Box Jumbling* and *Freeze* and will help in understanding how these models are utilizing temporal information. Sampling rate slows the playback speed by sampling frames uniformly at a varying level of rates and reverse samping does so in the reverse order of the original sequence. Jumbling splits a video into segments and randomly shuffles the frames in that segment while Box jumbling randomly shuffles the segments. Freezing simulates when live streaming buffers, freezing on random frames for random durations.

## 3.2 Text Perturbations

We group text perturbations into three different types, natural, machine-based, and synthetic. Here machine-based perturbations use a model to alter the text while natural-based imitates real-world mistakes when generating text. Synthetic are not natural but are used to gain a greater understanding of the models. The text perturbations are further grouped into seven different categories *ChangeChar*, *AddText*, *Bias*, *Positional*, *DropText*, *SwapText* and *TextStyle* with a total of 35 different perturbations. **ChangeChar** refers to any perturbation that changes a character in word(s). **SwapText** is a machine-learning based perturbation that swaps word(s) from the original phrase. **AddText** includes appending irrelevant phrases to text or inserting adverbs. **TextStyle** are perturbations that change the original text's style, e.g. making it *passive* [14]. **Bias** perturbations include switching the gender of word(s) in a phrase [47]. We additionally include changing all male references to female, the reverse, and convert all gender-specific references to gender neutral.

**DropText** perturbations are synthetic and drop words based on their part-of-speech (POS) tag. These perturbations are included to gain a better understanding of word level attention, more specifically, to understand if models attend more to objects, actions or context. *DropNN*, *DropVB*, and *DropVBNN* are different variations of dropping words based on whether the POS tags are Noun and/or Verb. Because there are often more nouns in a sentence, we have an additional perturbation *RandNN* where only one noun is dropped randomly as opposed to all. For example, "a little girl does gymnastics" becomes " a little [UNK] does gymnastics". In order to evaluate attention to contextual words, *OnlyNN*, *OnlyVB*, and *OnlyNNVB* drops all words but those with POS NN and/or VB. **Positional** perturbations are machine-based and alter the phrase based off their location. This is used to evaluate the models based on the position of words in a phrase. These include *DropFirst*, *DropLast*, *DropFirstandLast*, and *ShuffleOrder*. Drop-related perturbations will replace a word at that position with an [UNK] tag. The ShuffleOrder perturbation shuffles the words in a phrase randomly. More details on the generated text perturbation are provided in the Appendix.

# 4  Robustness Benchmarks and Evaluation

## 4.1  Model Variants

We perform our experiments on five different self-supervised video-language models which are based on CNN and Transformer architectures. The goal is to benchmark multiple pre-training approaches while simultaneously study the behavior of CNN and transformer based models for robustness in

text-to-video retrieval. Models were chosen based on whether they provided 1) a usable code base, 2) model weights, 3) and used text and video as their modalities.

We evaluate the most popular video-language approach MIL-NCE [38] which uses a CNN backbone and Word2Vec word embeddings with an MIL-NCE contrastive loss between text-video pairs. We further evaluate models and approaches that utilize visual features from [38] with further training and different self-supervised approaches. The more recent method VideoClip [63] is a transformer-based approach relying instead on BERT [18] for both text and video encodings with a similar but improved contrastive loss. COOT [25] similarly uses transformer-based encoders taking BERT text features and S3D visual features as input and includes cross-attention between the text and video features. Rather than a contrastive loss with negative pairing, COOT focuses on alignment between text and video alone. UniVL [35], is another transformer-based approach that uses a cross-encoding transformer in addition to separate encoders as their self-supervised objective. The final approach evaluated, FIT [5], combines image-based research with video. It uses only a small set of frames for a given clip which is encoded using a Visual Transformer (ViT) [21, 7]. They also pre-train with a different dataset that comprises of both images from CC3M [52] and video from their own proposed dataset, Web2Vid [5]. FIT uses a contrastive loss for video-text pairs and for text-video pairs with temporal curriculum learning. More details on these approaches are shown in Table 1.

## 4.2  Datasets

We use two video-language datasets for our experiments: MSRVTT [55] and YouCook2 [69]. **MSRVTT** is a video captioning dataset which consists of 10,000 clips with an average length of 10 seconds each. These videos show a variety of activities that can be organized into 20 categories. We follow JSFusion [67, 38, 63] which randomly samples 1K clip-text pairs as test data for evaluation. **YouCook2** is a task-oriented cooking dataset with 2000 long untrimmed videos from 89 cooking recipes. Each video is annotated with captions with provided temporal boundaries, allowing each video to be split into a set of clips. There are 3,305 test clip-text pairs from 457 videos for evaluation.

Captions in the MSRVTT and YouCook2 dataset are quite different. YouCook2 has no indication of gender with phrases comprising 2x more nouns compared to MSRVTT while MSRVTT has a more uniform distribution of words with an increased vocab size of 568 more unique words. Videos in MSRVTT and YouCook2 are also different where YouCook2 are long, complex activities split into clips with temporally bounded annotations. The test dataset will have multiple clips from the same video while all test clips in MSRVTT are from different videos. *This means the distributions between the two datasets are different and may result in different observations.*

We apply 90 visual perturbation to the test videos, 31 or 35 text perturbations to the captions, and 66 visual and text combined perturbations for creating robustness benchmarks **YouCook2-P** and **MSRVTT-P**. YouCook2-P does not have gender-related perturbations because of no reference to gender in their captioning, therefore only 31 text perturbations are used. MSRVTT-P consists of 90,000 videos and 35,000 captions resulting in 2,766,000 video-text pairs. YouCook2-P consists of 41,130 videos split into 301,500 clips and 103,850 captions, resulting in 9,266,100 clip-text pairs.

## 4.3  Tasks and Evaluation Metrics

We evaluate the performance of models on text-to-video retrieval using a retrieval rate R@K metric [38]. To measure robustness, we use two metrics; one for absolute retrieval drop and the other for relative retrieval drop [28, 53, 57, 33]. Given a trained classifier model $f$, we first compute retrieval $R_c^f$ on the clean test set. Next, we test this classifier on a perturbation $p$ and obtain retrieval $R_p^f$ for perturbation $p$. The absolute robustness $\gamma^a$ is computed for each perturbation $p$ as $\gamma_p^a = 1 - (R_c^f - R_p^f)/100$. For visual perturbations, the aggregated performance of a model can be obtained by averaging all severity levels to get $\gamma_p^a$ and over all perturbations to get $\gamma^a \pm \sigma$. For text perturbations, the aggregated performance of a model can be obtained by averaging across sub-types rather than severity. To take into account differing model performance on the clean dataset, we compute relative performance drop to measure models robustness. The relative robustness $\gamma^r$ is computed for each perturbation $p$ as $\gamma_p^r = 1 - (R_c^f - R_p^f)/R_c^f$ which is the difference normalized to the accuracy of the model on the test set without a perturbation. The robustness score will usually range from 0 to 1, where 0 indicates a model is not robust and 1 is where the model is entirely robust. A score greater than 1 indicates that the model's performance is better with the perturbation.

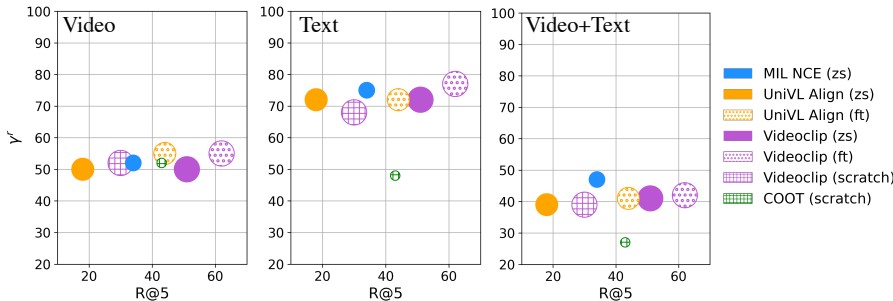

Figure 2: A comparison of models under different training protocols (zs: zero-shot learning, ft: finetuning on target dataset, and scratch: no pretraining) on thr YouCook3 dataset. The y-axis is the drop in performance when data is perturbed measured by the relative robustness $\gamma^r$ score, the x-axis is R@5 for text-to-video retrieval, and the size of marker represents number of model parameters. The three plots (left to right) corresponds to visual, text, and visual+text perturbations respectively.

Table 2: The drop in performance when data is perturbed measured by the Relative robustness $\gamma^r$ for each category of video perturbations on MSRVTT-P with SD $\pm\sigma$. The ViT based approach FIT is noticeably more robust on spatial noise as compared to the other approaches evaluated.

| Method | Blur | Camera | Digital | Noise | Temporal |
|---|---|---|---|---|---|
| FIT (scratch) | 0.67±0.13 | 0.90±0.12 | 0.84±0.07 | 0.73±0.24 | 1.00±0.01 |
| VideoClip (scratch) | 0.54±0.23 | 0.80±0.17 | 0.53±0.21 | 0.24±0.20 | 0.96±0.05 |
| FIT (zs) | **0.79±0.13** | **0.97±0.10** | **0.88±0.07** | **0.81±0.21** | **1.03±0.02** |
| MIL NCE (zs) | 0.59±0.17 | 0.78±0.11 | 0.42±0.20 | 0.24±0.19 | 0.88±0.06 |
| UniVL (zs) | 0.61±0.21 | 0.85±0.12 | 0.61±0.16 | 0.27±0.20 | 0.96±0.04 |
| VideoClip (zs) | 0.61±0.22 | 0.84±0.13 | 0.62±0.18 | 0.22±0.17 | 0.95±0.02 |
| FIT (ft) | 0.74±0.11 | 0.92±0.11 | 0.83±0.07 | 0.77±0.20 | 1.00±0.01 |
| UniVL (ft) | 0.60±0.19 | 0.85±0.12 | 0.58±0.19 | 0.27±0.22 | 0.90±0.09 |
| VideoClip (ft) | 0.59±0.23 | 0.84±0.13 | 0.62±0.19 | 0.26±0.21 | 0.95±0.04 |

## 4.4 Implementation Details

To ensure fairness to the original models, we use the official model implementations that were available with pre-trained weights with the same experimental setup as described in these works. These protocols vary between models and datasets. HowTo100M-MIL [38] take video as input and split the temporal boundary of the passed video into a clip of 4 with 32 frames for YouCook2 and 16 frames for MSRVTT. They take text as input and embed each word using Word2Vec. VideoClip [63] and COOT [25] use pre-extracted features from the pre-trained S3D-G [61] model provided by [38] while UniVL [35] uses pre-extracted features from the same model but before the final layer resulting in a smaller embedding size. VideoClip and UniVL take text as raw input while COOT [25] uses pre-extracted text features from BERT [18]. FIT [5] splits a clip into 4 segments and randomly selects 1 frame from each. These details are summarized in Table 1. We also analyze some models on whether they are fine-tuned, pre-trained or trained from scratch based on the availability of code. In the original implementations, VideoClip, Howto100-MIL and UniVL are pre-trained on HowTo100M [40], COOT was trained from scratch on MSRVTT, and FIT is pre-trained on CC3M [52] and Web2Vid [5]. Evaluating models using only pre-trained weights are considered *zero-shot* (ZS). FIT, VideoClip and UniVL were additionally *fine-tuned* (FT). Models that are trained on the evaluation datasets without pre-training are considered *scratch*.

## 5 Experiments

We perform our experiments with the studied models on YouCook2-P and MSRVTT-P benchmarks. A summarized overview of the robustness analysis of models against different perturbations on YouCook2-P is shown in Figure 2. Table 4 shows robustness scores aggregated across visual or real-world text perturbations for YouCook2-P and MSRVTT-P respectively. Table 2 show relative robustness scores aggregated across visual categories for MSRVTT-P. Table 3 shows relative robustness

Table 3: Relative robustness scores $\gamma^r$ with standard deviations $\pm\sigma$ for each category of distribution shifts for text perturbations.

| MSRVTT $\gamma^r$ | AddText | Bias | ChangeChar | DropText | Positional | SwapText | TextStyle |
|---|---|---|---|---|---|---|---|
| FIT (scratch) | 0.92±0.03 | 0.84±0.07 | 0.78±0.11 | 0.47±0.34 | 0.77±0.16 | 0.80±0.16 | 0.98±0.02 |
| VideoClip (scratch) | 0.90±0.06 | 0.88±0.05 | 0.78±0.09 | 0.46±0.32 | 0.73±0.13 | 0.81±0.18 | 0.96±0.02 |
| FIT (zs) | **1.00±0.00** | 0.96±0.04 | 0.79±0.14 | 0.53±0.36 | 0.84±0.13 | **0.87±0.18** | 1.01±0.02 |
| MIL NCE (zs) | 0.78±0.00 | 0.90±0.03 | 0.77±0.10 | **0.57±0.32** | 0.78±0.15 | 0.75±0.12 | 0.91±0.02 |
| UniVL (zs) | 0.92±0.10 | **0.97±0.04** | 0.71±0.11 | 0.33±0.27 | 0.64±0.15 | 0.84±0.15 | 0.90±0.07 |
| VideoClip (zs) | 0.89±0.07 | 0.94±0.05 | 0.71±0.11 | 0.39±0.27 | 0.62±0.17 | 0.81±0.13 | 0.97±0.03 |
| FIT (ft) | 0.94±0.04 | 0.88±0.05 | 0.79±0.11 | 0.49±0.34 | 0.80±0.14 | 0.82±0.17 | 0.97±0.02 |
| UniVL (ft) | 0.92±0.05 | 0.88±0.05 | 0.80±0.09 | 0.49±0.31 | 0.78±0.13 | 0.81±0.14 | 0.96±0.01 |
| VideoClip (ft) | 0.94±0.04 | 0.91±0.05 | **0.81±0.09** | 0.53±0.32 | **0.87±0.08** | 0.83±0.15 | 0.97±0.02 |
| **YouCook2 $\gamma^r$** | AddText | Bias | ChangeChar | DropText | Positional | SwapText | TextStyle |
| COOT (scratch) | 0.88±0.12 | — | 0.18±0.29 | 0.41±0.37 | 0.76±0.12 | 0.51±0.43 | 0.57±0.51 |
| VideoClip (scratch) | 0.85±0.12 | — | 0.62±0.13 | 0.37±0.33 | 0.69±0.11 | 0.72±0.18 | 0.92±0.05 |
| MIL NCE (zs) | 0.92±0.03 | — | 0.74±0.14 | **0.57±0.39** | **0.83±0.15** | 0.75±0.18 | 0.98±0.01 |
| UniVL (zs) | **1.14±0.03** | — | 0.75±0.10 | 0.43±0.41 | 0.80±0.24 | 0.75±0.17 | 0.94±0.09 |
| VideoClip (zs) | 0.95±0.04 | — | 0.77±0.10 | 0.47±0.33 | 0.70±0.13 | 0.77±0.14 | 0.94±0.04 |
| UniVL (ft) | 0.91±0.08 | — | 0.74±0.09 | 0.45±0.33 | 0.76±0.07 | 0.78±0.15 | 0.95±0.02 |
| VideoClip (ft) | 0.95±0.03 | — | **0.84±0.10** | 0.50±0.35 | 0.82±0.09 | **0.81±0.18** | **0.99±0.07** |

Table 4: The aggregated performance measured by Relative Robustness $\gamma^r$ and Absolute robustness scores $\gamma^a$ across model and training procedure with standard deviations $\pm\sigma$. For text, we aggregated only natural distribution shifts, excluding Positional and DropText perturbations.

| Method | MSRVTT-P | | | | YouCook2-P | | | |
|---|---|---|---|---|---|---|---|---|
| | Video | | Text | | Video | | Text | |
| | $\gamma^a$ | $\gamma^r$ | $\gamma^a$ | $\gamma^r$ | $\gamma^a$ | $\gamma^r$ | $\gamma^a$ | $\gamma^r$ |
| COOT (scratch) | — | — | — | — | 0.79±0.16 | 0.52±0.36 | 0.75±0.19 | 0.44±0.44 |
| FIT (scratch) | 0.93±0.08 | 0.84±0.18 | 0.94±0.05 | 0.87±0.11 | — | — | — | — |
| VideoClip (scratch) | 0.83±0.15 | 0.63±0.32 | 0.94±0.04 | 0.87±0.10 | 0.86±0.11 | 0.53±0.35 | 0.95±0.05 | 0.83±0.18 |
| FIT (zs) | **0.96±0.06** | **0.91±0.15** | **0.97±0.05** | **0.92±0.13** | — | — | — | — |
| MIL NCE (zs) | 0.89±0.08 | 0.60±0.29 | 0.96±0.02 | 0.85±0.09 | 0.84±0.13 | 0.53±0.37 | 0.95±0.05 | 0.86±0.15 |
| UniVL (zs) | 0.94±0.05 | 0.67±0.30 | **0.97±0.02** | 0.85±0.14 | **0.91±0.07** | 0.50±0.36 | **0.98±0.03** | 0.88±0.17 |
| VideoClip (zs) | 0.92±0.07 | 0.66±0.32 | **0.97±0.03** | 0.86±0.14 | 0.74±0.19 | 0.50±0.37 | 0.93±0.06 | 0.86±0.11 |
| FIT (ft) | 0.92±0.09 | 0.86±0.15 | 0.93±0.06 | 0.88±0.10 | — | — | — | — |
| UniVL (ft) | 0.82±0.15 | 0.65±0.29 | 0.94±0.05 | 0.88±0.09 | 0.80±0.16 | **0.55±0.36** | 0.93±0.05 | 0.85±0.12 |
| VideoClip (ft) | 0.82±0.16 | 0.66±0.30 | 0.94±0.05 | 0.89±0.09 | 0.72±0.23 | **0.55±0.37** | 0.95±0.06 | **0.91±0.10** |

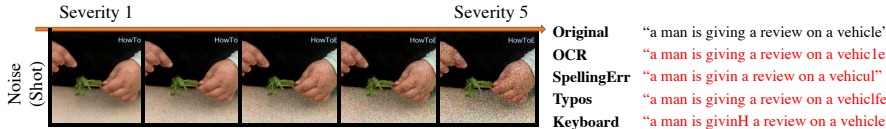

Figure 3: Examples of perturbations that humans are able to perceive but models struggle with.

scores across different text categories for both datasets. More detailed results, including a breakdown of each perturbation category, are provided in the Supplementary. Next, we provide more insights and analysis on different interesting observations in this study.

**Training Strategy**  Table 4 split models by their training strategy. These results indicate that for MSRVTT-P, models that are zero-shot are typically higher in absolute and relative robustness. For long, complex activities in YouCook2-P, fine-tuned models are typically more relatively robust. Pre-training data choice may also play a factor. FIT pre-trains on both images and video as opposed to the majority of the other approaches that pre-train or use features pre-trained on the HowTo100M dataset [39]. While FIT performs well on MSRVTT, when zero-shot evaluating FIT on YouCook2 without perturbations, the results are an R@5 of 7.5%, indicating this may only be the case for short activity videos like in MSRVTT. *In summary, pre-training models typically improves both performance and robustness against real-world and synthetic distribution shifts.*

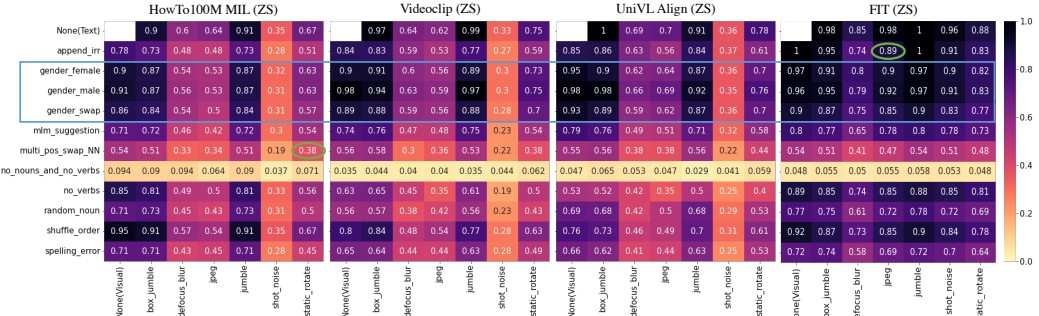

Figure 4: The drop from R@5 performance on clean to when perturbed by combinations of text and visual perturbations measured by Relative Robustness $\gamma^r$. The x-axis shows text perturbation and the y-axis shows visual perturbation. The first row/column show scores for the respective text/video perturbation when not combined. An example of compounding effect is circled in green. Gender perturbations are boxed in blue.

**Human Perceivable Perturbations**   *Noise* and *Blur* are pixel-based visual perturbations which humans can easily filter (Fig. 3). These perturbations are also ones models are least robust as shown for MSRVTT-P in Table 2. While the models pre-trained on HowTo100M using a CNN backbone feature extractor perform poorly on spatial noise in MSRVTT-P, the ViT based approach FIT is more relatively robust to noise. On text perturbations for both datasets shown in Table 3, **between the semantic preserving** text perturbations, models are **least robust to ChangeChar**, indicating that text models are still unable to recognize small changes that humans will perceive in text (Fig. 3). *This indicates that visual-language models are **not typically robust to real-world distribution shifts that are human perceivable** such as character changes in word(s) and additive noise.*

**Architecture**   There are typically two architecture types for video-language models, a two-branch or cross-attention encoder. Two-branch encoders keep the visual and text encoders separate with the only interaction being the propagated loss. Cross-attention utilizes a form of cross-attention between a visual and language encoder before calculating loss. Most models use a two-branch encoder approach as they find it performs better [63]. However, of the models we studied and evaluated on YouCook2-P, COOT [25] and UniVL [35] use cross-attention while VideoClip [63] and MIL NCE [40] use two-branch. Looking at Table 4, UniVL typically has higher absolute robustness scores compared to the two-branch encoder based approaches. *Based on the models studied, this indicates the cross-attention may improve performance on long, complex activities with little cost to robustness.*

The visual encoder architecture also varies for the different approaches. The majority of the models studied here use a 3D CNN for video feature extraction that is input into a transformer. However, the FIT [5] model uses a small set of frames input to a ViT. When looking at Table 2, there is a noticeable relative robustness difference between FIT and the other approaches. This indicates that ViT transformers may be more relatively robust than CNN based approaches. However, the FIT zero-shot model does not perform well on YouCook2, with a baseline R@5 of 7.5%, indicating that *using a ViT may be highly robust against short activities but not necessarily long, complex activities.*

Text encoders also vary across models. However, almost all approaches utilize a BERT [18] transformer while only MIL NCE [40] use a Word2Vec [41]. When text is perturbed on *DropText*, *Positional* and *ChangeChar*, Word2Vec is more robust than BERT on zero-shot evaluation (see Figure 5). *Based on the models studied, these results may indicate that when using keywords as opposed to sentence descriptions, Word2Vec may be a more robust approach compared to BERT.* Additionally, as shown in Figure 2, COOT's relative robustness is noticeably worse when text is perturbed. Because COOT uses pre-extracted text features without any training, this may indicate that using pre-extracted text features is not a robust method due to learning pulling the video feature space towards the existing text space rather than pulling both video and text closer together in a new, joint space.

**MultiModal Perturbations**   To understand the compounding effects of shifting distributions in both the visual and text domain, we select a subset from each perturbation with at least one from each

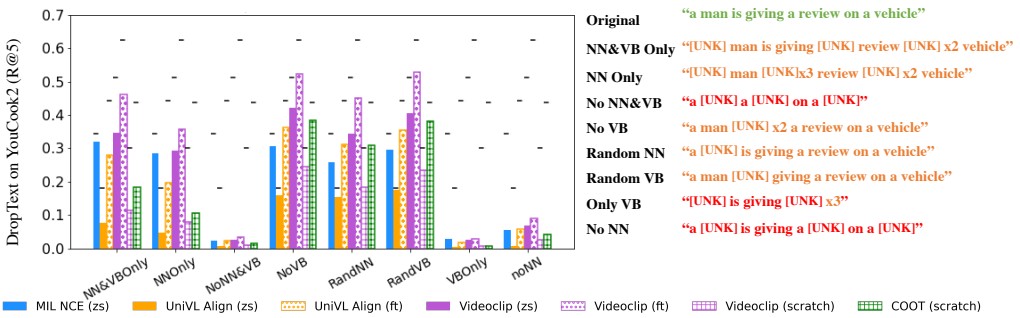

Figure 5: Performance for *DropText* perturbations on YouCook2-P. Dashes are R@5 on clean and bars are R@5 on perturbed. Examples of these perturbations are provided: red is where models struggle the most and orange indicates models are surprisingly robust.

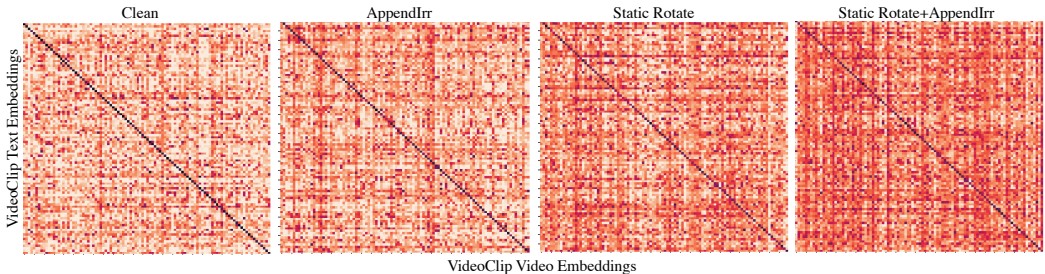

Figure 6: Similarity matrices where the x-axis are video representations and the y-axis are text representations sampled from VideoClip on the YouCook2 dataset. The darker the color, the more similar. When both video and text are perturbed, a compounding effect is shown by the increase in similarity for samples that do not match.

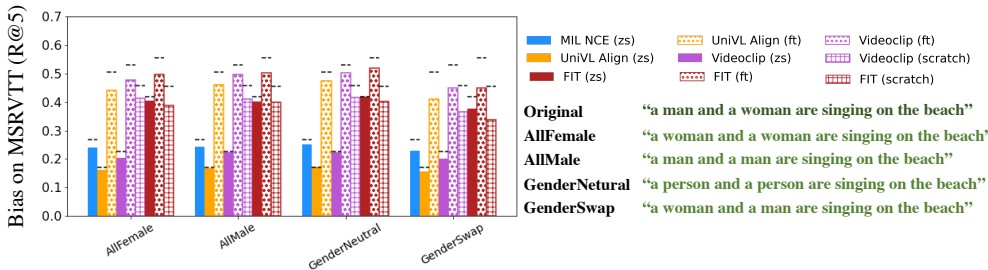

Figure 7: Performance on *Bias* perturbations on MSRVTT. Dashes are R@5 on clean and bars are R@5 on perturbed. Models are less robust when gender is swapped or male/female terms are changed.

higher-level category. For visual perturbations, we use a severity of 3. Figure 4 shows a summary of these results on the MSRVTT dataset. There are certain combinations of perturbations that are more challenging for models as compared to others. For example with FIT (ZS), the model has a relative robustness score $\gamma^r = 1$ for *AppendIrr* and $\gamma^r = 0.98$ for *JPEG* compression, but when combined, the score is $\gamma^r = 0.89$ (see Figure 6).

Meanwhile, some perturbation combinations will be close to the lowest $\gamma^r$ between the two, e.g. no nouns and no verbs and shot noise. Even when a model is equally robust to perturbations in isolation (e.g. Jumble and GenderMale on HowTo100m MIL), there is a decrease in overall robustness when combined. In summary, *when both text and video are perturbed, models are less robust than when the same perturbations are applied in isolation, with some combinations worse than others.*

**Bias** To evaluate bias in models, we evaluate the robustness to gender-specific changes to text on the MSRVTT dataset. In the MSRVTT dataset, the most common part-of-speech (POS) tagged nouns were "man" and "woman" with "man" references 2x that of "woman". When the original text was perturbed, 33.8% of male references were converted to female, 24.3% of female references were converted to male, 53.2% of phrases swapped gender and finally 52.8% of gender references were made neutral. Figure 7 visualizes these results where the dotted, horizontal line is the original text-to-video retrieval score and the bar are the new scores with the perturbed version of text. *The results indicate that models are **less robust when the gender is all female** and when the gender is swapped from male-to-female and vice versa.*

**Temporal** Temporal perturbations are used to evaluate whether models use temporal information or not. Figure 8 visualizes the results of these experiments. Models show strong robustness to the video-specific temporal perturbations *Jumble*, *Sampling*, and *Freeze* (a breakdown of robustness scores is provided in the Supplementary). This indicates that fine-grained temporal elements are not necessarily important to these video-language models. This also indicates that the activities do not necessarily change when in reverse.

On YouCook2-P, which consists of untrimmed, minutes-long videos, none of the models are robust to *BoxJumble*. This indicates that the models require alignment between the visual ques and the respective text, but temporal order within the aligned segment is not utilized. While this shows poor robustness for this perturbation, it shows good model behavior. *These results indicate that both visual and textual cues are used during learning however the **models are attending more to objects and scene rather than motion and activity**.* This is similar to how humans may describe different videos where nouns and descriptors are more differentiating as opposed to activities which could describe a group of videos.

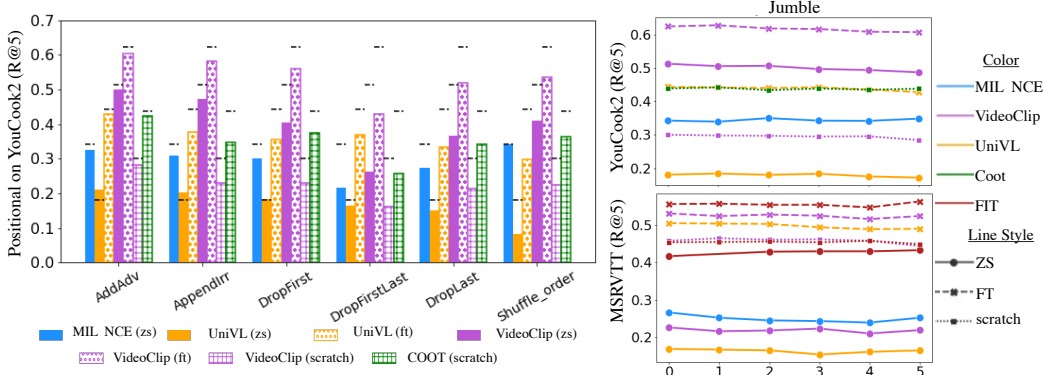

Figure 8: Temporal perturbations for text (left) and video (right). On text, dashes are R@5 on clean and bars are R@5 on perturbed. Models appear to be typically robust, especially MIL NCE for *ShuffleOrder*. For video, the x-axis is the severity where a severity of 0 is performance on clean and y-axis is R@5. Models show little change in performance, indicating that temporal order is not utilized in these approaches.

## 6  Conclusion

In this work we propose a robustness benchmark for video-language models and provide initial insights of several multimodal approaches. In order to perform this study, we create two benchmark datasets, MSRVTT-P and YouCook2-P. Our empirical study provides several interesting insights into the behavior of some of the existing models on the proposed benchmarks. Some key observations are 1) models are generally more robust when only text is perturbed as opposed to when only video is perturbed, 2) models that are pre-trained are typically more robust with improved performance on zero-shot evaluation 3) models attend more to scene and objects rather than to motion and action. The findings and the benchmark in this work can potentially open up interesting future research on robustness of video-language learning.

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
