# OpenReview forum: "Robustness Analysis of Video-Language Models Against Visual and Language Perturbations"
_NeurIPS.cc/2022/Track/Datasets_and_Benchmarks — NeurIPS 2022 Datasets and Benchmarks _

### Official Review · Reviewer_ZxJh · 2022-07-19

**Rating:** 6
**Confidence:** 4

**Strengths:**

The paper addresses an important topic of evaluating multi-modal models with regards to their robustness against perturbations.

The paper is well-written and easy to understand.

The paper evaluates a set of common pretrained models and provides some overarching results that are informative for developing multi-modal models that are robust.

**Weaknesses:**

## Temporal perturbations for video models.
I focus on this temporal aspect because it is, in my opinion the most important one that contrasts prior robustness work such as that from Geirhos et al. to this work. Therefore getting this right is crucial for providing a contribution to the community.

### Effect of perturbation might well be dataset dependent.
E.g. if a caption reads "a man jumping from left to right", changing the temporal order might well change the semantics of the video. This might be more acute for datasets like SomethingSomething, but an analysis of these actions which are asymmetric with time would be needed for this to indeed properly measure robustness.
A good reference here might also be the Arrow of Time paper by Wei et al. (CVPR 2018).

### Do we instead _want_ temporal un-robustness?
With methods such as CLIP working extremely well on solving many tasks simply by looking at a single frame, should strong video-models (that truly look at motion instead of scenes) be un-robust instead? It would therefore be very interesting to analyse the robustness (and downstream) performances against this temporal robustness to see if there are patterns.
Additionally, the work from Buch et al (CVPR 2022) might provide additional information about where the temporal aspects are indeed necessary.

### Finer grained analysis of temporal perturbations.
As outlined above, temporal reversal might very well make perfect sense for many videos (e.g. "ball bouncing"), while not for others (e.g. "car crashing"). Therefore mixing the robustness results of this experiments with other perturbations such as frame shuffling might blur some informative signal that could be extracted from the experiments.

## What about more recent multi-modal works.
That are trained on HT100M like GDT (Patrick et al. ICCV 2021, FiT (Bain et al. ICCV 2021), BraVe (Recasens et al. ICCV 2021)? For the first two works, models are also publicly available I believe and are also relevant related work. I believe running further evaluations should not take too much effort but cementing the results with further evidence (especially with more recent models -- the MIL-NCE model the authors use is from 2020), would strengthen the paper.

**Additional Feedback:**

I have included the points that I would like to have a response to in the weaknesses section. I believe addressing these points would make for a stronger and overall better rounded paper and am willing to change my rating based on the replies.

**Clarity:**

The paper is written well, although some Fig 4 and 5 are very convoluted and hard to grasp.

**Correctness:**

The benchmark and its evaluation seems to be correct, apart from the possible over-aggregation of temporal perturbation results mentioned in the weaknesses section.

**Documentation:**

I would urge the reviewers to also provide a datasheet for the dataset (Gebru et al. CACM 2021). While the NeurIPS checklist is the bare-minimum, providing a full datasheet would be more informative and also provide thorough information about maintenance and copyright and hosting. What happens if MSRVTT (or YouCook) gets removed etc.?

**Ethics:**

There are no ethical concerns.

**Relation To Prior Work:**

Prior work is well mentioned, as far as the reviewer can tell.

**Summary And Contributions:**

This paper proposes a new benchmark for video-text robustness evaluation. For this, the authors propose to reuse the popular MSRVTT and YouCook2 benchmarks by applying many kinds of noises to and evaluating models by their text-video retrieval and VideoQA performances.

---

### Official Review · Reviewer_bexS · 2022-07-21
**Review for Paper 86**

**Rating:** 4
**Confidence:** 4

**Strengths:**

- The paper approaches an important topic on the robustness of multimodal models. This is the first paper that tests the robustness on video-text cases. This is an interesting topic for the community since it can also help understand the relevance of each modality and the robustness of methods when the train and test dataset distributions are mismatched for either modality.

- The variation of the different perturbations proposed is wide and could potentially give an idea of the robustness of the methods on real-world perturbations.

- The models used for the study are relevant in the field.

- The analysis based on type of perturbation, model type, and training scheme is an interesting way to compare the robustness of models under different circumstances.

**Weaknesses:**

1. The paper states 3 different contributions.
Contribution 1 ("analyze the robustness of multimodal models against different real-world distribution shifts.") and 3 (Provide insights including comparison of different model architectures, training procedures and effect of various perturbations on model performance.) corresponds to the same contribution, i.e. provide an analysis of the robustness of 4 different models on two video-text datasets under several perturbations and different training schemes. This should be stated as a single contribution.

2. The paper claims to study the effect of "real-world perturbations" on multimodal models.
In what real case a video would contain a "Jumbling" or "Box Jumbling" perturbation? To my intuition, these perturbations would show the relevance of the temporal information for selected models, but are not found in the real-world.
The "DropText" replaces words with [UNK] tokens which creates captions that do not appear in the real-world. Additionally, I question the relevance of removing all nouns and verbs from the text. This perturbation destroys all semantic information in the captions leading to a wrong matching with any video.

3. There is a typo in line 197 with the subscript of the retrieval score. The absolute robustness metric is defined in line 199, while the relative robustness \gamma^r used for most of the tables is not defined in the paper.
The absolute robustness is only used in table 5 for the VideoQA in the main paper.
Given that the relative robustness definition is not presented, it is hard to make conclusions from the scores presented in the tables.
The absolute robustness score does not, as the name states, show the significance of the drop with respect to the original performance of every method.
For example, if the original clean score is low, a small drop in performance (high absolute robustness) due to perturbations would represent a low relative robustness. Absolute and relative metrics should be used together when comparing robustness.

4. Based on Figure 2, the overall robustness comparison with respect to visual, text, and both modalities perturbations, why do you think Coot is not robust to text perturbations but more robust when both video and text are affected?

5. The paper has several claims in the summaries of each section ( "a two-branch approach is typically more robust (...)", "Word2Vec is more robust on semantic-changing (...) BERT is more robust for non-semantic changing perturbations (…)"). However, to make these claims it is required to test a bigger sample of methods, or more generic and equally trained methods, than just one which uses Word2Vec vs 3 BERT-based methods, and one cross-attention vs 3 two-branch approaches.

6. Figures 5, 6 and 7 are very hard to understand. The black lines are not well explained, only after line 293 one can know they correspond to the clean scores, which are necessary for understanding the figures. Plots are missing axis titles and explanation of the content. As a general remark, the presentation of the figures needs to be improved.

**Additional Feedback:**

The topic or robustness in multimodal models is relevant to be studied and the dataset with perturbations looks promising. My main concerns are more regarding the metric and the generalization of claims with a small sample of models. I would be interested in this work once these concerns are addressed.


**Clarity:**

The paper needs work on the clarity of the statements and more on the clarity of tables and figures. Including axes titles and ranges of colors. A key advice is to show a single idea per table. Refer to weaknesses 3. and 6.

**Correctness:**


My main concerns are two:
- Definition, clarity, and relevance of the metrics used to compare the methods. Refer to weakness (3).
Without a clear way to compare the methods, all arguments and statements cannot be verified.

- General statements are made with a very small sample of models under representing the possibilities with training schemes, architecture design and embedding models. For example, including more models which use Word2vec and cross-attention modules would bring more significance to the results.


**Documentation:**

Yes

**Relation To Prior Work:**

Yes

**Summary And Contributions:**


The paper brings up a study on the robustness of multimodal models which uses text and video. It proposes a benchmark of video-caption pairs with different types and intensities of perturbations for each of the modalities. This benchmark is built using two well known video-caption datasets by adding perturbations on the two modalities and evaluating on the perturbed sets. The paper compares four different models and uses the text-to-video retrieval score to calculate the robustness of these models depending on the perturbation. This work proposes a metric to measure the robustness based on the retrieval scores.

---

### Official Review · Reviewer_GJKq · 2022-07-21
**This is a nice paper, accepted**

**Rating:** 8
**Confidence:** 3
**Correctness:** 1)	i think dark color representing bi…

**Strengths:**

1）originality：The submission is of full oringinality. It  points out a novel but important and realistic idea. To find out the robustess against realistical distribution shift, the submission perform lots of experiments and create two new datasets and some  perturbation.
2）Quality：the submission is high of quality. The team performed enough of experiments, adopt reasonable metrics to evaluate the results.
3）Clarity：the organization of the paper is nice. For example, the abstract and introduction are simply but clearly clarifing the shortness of current research of multi-modal and the significance of their work. And the other parts of the paper represents their workload very well.
4）Significance：as mentioned in the originality,the submission not only perform the first extensive experiment in the robustness of multi-modal approach against realiatic distribution shift, but also create two related datasets and define several perturbations, which will contribute to the future related research.


**Weaknesses:**

1)There are some problems about the figure.the figure 4,as far as i am concerned, I prefer dark color representing big value to plain color representing big value, because i think the former is more intuitive.
2)the figure 6 looks like embedded into the text, and it influences the overall layout of the paper.
3)table 5 has the same problems as figure 6. And table5 lacks the description.


**Additional Feedback:**

1.I think the author can change the style of insert figure and table. For example, when inserting a figure or a table, don’t embed them into the text, because it will influence the overall look and feel of the paper.I think starting a new line for each figure and table is a good choice.
2.I think the author can add more previous work to the part of related work. The part of related work of the submisssion is too short to explain the basis of the submission.


**Clarity:**

The paper is well written. For example, the abstract and introduction are simply but clearly clarifing the shortness of current research of multi-modal and the significance of their work. And the distribution shift part and model variants explain elarly. The Robustness Benchmarks and Evaluation clearly explains the metrics and the implementaion details. The experiments part show their experments setting and the results clearly.

**Documentation:**

The authors explain the organization, implementation and the contents of their new datasets but didn’t make it availability yet. I think it might be a pity because their datasets may provide big convenience to future related work.
In terms of benchmarks, i think the author has provided enough details to implement, so i think the benchmark is nice.


**Ethics:**

As far as I am concerned, there is no ethical concerns.

**Relation To Prior Work:**

The submission clearly discussed how this work differs from previous contributions. The author explain their perturbations in section3.2 and the new datasets in section 5.1. i think the content of these two parts is clearly enough to explain their difference.

**Summary And Contributions:**

The submission show a fact that the robustness of some approaches of multi-modal against real-world perturbations has not been  studied and they perform the first robustness study about the aspect. After experiments, they found some models and encoders are more robust on some conditions. I think these contribution will act as benchmark for the future research in the robust multi-modal learning.

---

### Official Review · Reviewer_oYwi · 2022-07-26
**A benchmark on multimodal robustness in text-to-video retrieval**

**Rating:** 5
**Confidence:** 5

**Strengths:**

1. The paper provides organized and detailed descriptions of different visual and text perturbations applied in creating the benchmark and the types of perturbations are relatively comprehensive.
2. The choice of SOTA baselines is properly justified with detailed descriptions and comparisons of model architectures, objectives, and training strategies.
3. Empirical results are carefully analyzed with highlighted conclusions, which are easy to follow.

**Weaknesses:**

Major concerns from the reviewer include:
1. The scope of multimodal robustness evaluation provided by the two proposed datasets is fairly limited (only evaluated on the text-to-video retrieval task, with very little mention of VideoQA towards the end of the paper). Furthermore, given this evaluation scope, either the contribution is limited or the generalization of the conclusions becomes less convincing.
2. The proposed metrics are not novel, and the reviewer believes they are highly similar to the proposed metrics in the following paper
> Liang et. al. *MultiBench: Multiscale Benchmarks for Multimodal Representation Learning.* NeurIPS 2021 Datasets and Benchmarks Track.

(which does not appear in the related work and the paper should clarify the different contribution) - note that the absolute robustness metric is the same while the reviewer does not find any formal definition of the relative robustness metric throughout the paper

3. Only observations but their interpretations are provided in the paper. It will be more interesting to provide more insights into the key observations that are less intuitive. For example, how do cross-attention and pre-training relate such that cross-attention and the two-branch approach are equally robust in the pre-trained case? Moreover, since both cross-attention and pre-training affect model robustness, does the comparison experiment on the claim that pre-training helps robustness have any confounding factor?
4. What real-world distribution shifts will the temporal perturbations (in particular, Box Jumbling) correspond to? The drop in performance caused by random shuffling of frame segments sounds reasonable to models that utilize more temporal dependencies instead of a robustness issue.
5. Please consider fixing typos (e.g. line 176) and confusing notations (what is $i$ in the subscript, line 197).

**Additional Feedback:**

No.

**Clarity:**

Considering a large number of types of perturbations presented, the paper is organized and relatively easy to follow except for a few typos.

**Correctness:**

The dataset is constructed properly, but there are a few concerns regarding potential extra confounders of the experiments and the generalization of the claims made in the paper, which are stated in Weaknesses 1 and 3.

**Documentation:**

The reviewer does not find relevant codes that compute the robustness measures proposed in the paper. Since robustness evaluation makes up a key part of the paper's main contribution, it will facilitate relevant research in the community with evaluation codes provided by the official repo.

**Ethics:**

No.

**Relation To Prior Work:**

The reviewer cannot tell a significant contribution made by the paper in addition to the following paper:
> Liang et. al. *MultiBench: Multiscale Benchmarks for Multimodal Representation Learning.* NeurIPS 2021 Datasets and Benchmarks Track.

while the listed paper serves a more general purpose of evaluation.


**Summary And Contributions:**

This paper proposed two large-scale benchmark datasets by applying 90 visual and 35 different textual perturbations to the existing retrieval datasets MSRVTT and YouCook2. The benchmark is evaluated using several SOTA baselines of different input features and training strategies and two proposed metrics. The evaluation is followed by a comprehensive and careful analysis of the robustness of multimodal models under real-world distribution shifts and several interesting observations on how pre-training, architecture, etc. affect multimodal robustness.

---

### Official Review · Reviewer_TDGL · 2022-07-26
**Comments on Multi-modal robustness**

**Rating:** 7
**Confidence:** 4
**Correctness:** The method looks sound, aside from th…
**Clarity:** Yes, the paper is clear.

**Strengths:**

1. The paper proposes a multi-modal benchmark with two large-scale datasets, which is useful for research in both modalities as well as multi-modality.
1. The perturbation category is clear and comprehensive. All types of perturbation strategies are concisely summarized.
1. The paper points out the weakness in several multi-modal models and draws some conclusions useful for model design.

**Weaknesses:**

1. The evaluation metrics are not clearly explained. Many terms used in the paper are not clear. For example, in line 196, absolute retrieval drop and relative retrieval drop are not introduced. In Tables 2, 3, and 4, the meaning of the relative robustness score is not explained. It's better to introduce the metrics before using them in the experiments.
1. Some conclusions do not match the experimental results. Correct me if I'm wrong. For example, I don't think the conclusion on line 228 (fine-tuned models are slightly more robust on machine-based and natural test perturbation while being 6-8% more robust on synthetic perturbations) is correct based on the results.
1. Some captions do not match the numbers in the table. For example, in Table 4, two-branch encoders are only more robust on YouCook2 dataset while being less robust on MSRVTT dataset.
1. The Figure does not match the introductions in the paper. In appendix Figure 1(b), some perturbations from ChangeChar should belong to machine-based text perturbations. Also, in the paper, we have the synthetic type while in the figure we only have natural and machine-based types.

**Additional Feedback:**

See weakness

**Documentation:**

Yes, the code is provided.

**Ethics:**

No ethical concerns.

**Relation To Prior Work:**

Yes, it's well-discussed.

**Summary And Contributions:**

The paper proposes two text-to-video retrieval datasets and conducts comprehensive robustness experiments to study the robustness of multi-modal models against distribution shifts. The authors consider 90 visual and 35 textual perturbations in total. The paper also draws some interesting conclusions based on the experiments under different settings

---

### Official Review · Reviewer_f6NK · 2022-07-28
**A novel and useful benchmark for multimodal robustness that just needs some more precise language.**

**Rating:** 7
**Confidence:** 4

**Strengths:**

This work provides a formulaic and well-reasoned approach to creating a suite of tests to benchmark multimodal video-text models. The perturbations are both numerous and common sense offering for some very granular analysis if desired. I can easily see this becoming a common-place task to benchmark future models similar to the spirit of ImageNetv2 for classification robustness. The broader community would definitely enjoy the utility provided by the dataset.

Since it is built from popular existing datasets, the framework is extensible and likely to be maintained for the foreseeable future. It poses no ethical risks.

**Weaknesses:**

## Major

While the paper gives a helpful early analysis of the existing multimodal models, it struggles to give the proper nuance to some of its conclusions. Although the appendix includes a section on limitations, it focuses more on future directions that are imperative to multimodal understanding rather than inherent limitations of the work and of its methods.

The exploration gives some very interesting initial findings and trends that could have important impact on how we design models in the future. However, the limited number of models studied (N=4 unique architectures) should strongly bound the claims made.

## Minor

What does the $\pm$ mean in the tables? Standard deviation? 95% confidence interval?

While absolute robustness $\gamma^a$ is defined, relative robustness $\gamma^r$ is not. What does it signify?

While helpful, the figures are challenging to read at times. In Figure 2 the pairing of black on purple makes the textures nearly impossible to read. Are there alternative textures/colorings that can be used?

What are the floating dashes in Figure 5? What is the y-axis representing in figures 5 and 7? Could it be so that the text on the right side of Figure 5 matches the ordering of that on the x-axis?

In line 138 "**SwapText** and is any" should become "**SwapText** is a."

In line 266, if "significantly" is not referring to the statistical usage, I recommend a different word.

**Additional Feedback:**

I'd be happy to increase my score if the authors can provide further experiments to support their claims or a plan describing how they will hedge their language as well as revised figures.

The core contributions of the work are solid. As is it needs a little bit of cleaning up, but I can see it as a very helpful tool to quantify future multimodal robustness.

**Clarity:**

The paper's core elements are easy to follow and come with visual examples. It reads well and makes sense. It has some minor issues like a missing definition and the understanding of the figures. Section 6 goes a tad overboard with the use of italics and bolding. They help to emphasize the important pieces of information, but they are used a little too frequently. Overall, the paper's contribution can be coherently understood and interpreted.

**Correctness:**

The dataset is constructed in a fair and sound way. The metrics and perturbations make sense with the task at hand.

Claims made by the work are consistent with the finding in the data, but are severely lacking in their understanding of limitations. There is not enough evidence to make the claims as strongly as they are being made. For example we can not make the conclusions discussed about Word2Vec if only one model was trained on it. We can say there are initial trends that may lead to this conlusion, but as is it is not a fair conclusion.

**Documentation:**

All aspects are clearly defined. There is a well-documented GitHub repository that can automatically create the perturbations mentioned in the paper. It may be both reproduced and built upon.

**Ethics:**

The work poses no ethical concerns.

**Relation To Prior Work:**

The work does a good job at contextualizing multimodality and the study of robustness.

**Summary And Contributions:**

The benchmark provides the first two datasets to examine multimodal robustness in video-text settings. Through algorithmic perturbations to YouCook2 and MSRVTT, the work offers a lens to understand how both textual and visual alterations affect retrieval. After introducing the benchmark, the work continues to explain how existing multimodal models perform on the task and analyzes early trends that may be caused by the modeling choices.

---

### Comment · Area_Chair_gPDL · 2022-08-22
**Thoughts after reading other reviews?**

Dear Reviewers,

It appears there was no author response, unfortunately. We do have some diverging reviews here. Please read each other's reviews and see if your opinion changes. Thanks!

Best,
AC

---

### Meta-Review · Area_Chair_gPDL · 2022-09-10

**Recommendation:** Accept
**Confidence:** 5

**Metareview:**

This paper proposes new benchmarks for probing video-text models' robustness, that include a multitude of visual/textual perturbations. All of the reviewers have acknowledged the usefulness and effort put into the benchmark construction and presented analysis. Some of their concerns centered around: lacking metric definitions, limited scope (video-text), detachment from real-world scenarios, issues with figure/table presentation, etc. After rather extensive discussions, many of the concerns have been resolved. Presently, 4 out of 6 reviewers argue for acceptance (3 of them strongly). The remaining 2 reviewers maintain their opinion on the limitations of the presented work, most importantly somewhat inconclusive takeaways, limited number of models compared (initially 5, the authors have added one more), mostly one task (video-text retrieval). I believe having 6 models is acceptable for the proposed study. Upon carefully examining the claims/arguments, I encourage the authors to "scale down" their claims/narrative (and perhaps even rename the paper) to **more explicitly acknowledge the emphasis on video-text retrieval**. But I still think the analysis as such is valuable and thus recommend acceptance.

---

### Decision · Program_Chairs · 2022-09-16

Accept